# OpenReview forum: "At the Edge of Understanding: Sparse Autoencoders Trace The Limits of Transformer Generalization"
_ICML.cc/2026/Conference — ICML 2026 regular_

### Official Review · Reviewer_gcQj · 2026-02-26

**Soundness:** 3
**Presentation:** 2
**Significance:** 3
**Originality:** 3
**Overall Recommendation:** 4
**Confidence:** 4

**Summary:**

This paper treats SAE reconstruction as a probe of how “on-manifold” an input’s internal representation is, proposing that distribution shift forces the model to recruit more spurious concepts and yields higher SAE reconstruction error (Sec.3–4). The authors define an SAE-based “energy” score that combines reconstruction error with sparsity/$L_0$ activity (Sec.3.4–3.5; $F(x)$), and use it to characterize and detect OOD behavior. In a controlled setting, they induce OOD via increasing rates of typos and show that (i) task accuracy degrades while (ii) the number of activated SAE features increases (Fig.2), with results averaged across multiple random typo configurations. They then turn the signal into tools: thresholding the energy score yields strong typo OOD classification accuracy (Appendix: Table 5), the score supports sample-efficient fine-tuning via decile-based curriculum compared against entropy/Mahalanobis-style signals (Sec.5 / Appendix), and SAE-feature-space analysis is used to reduce jailbreak success via LoRA-style fine-tuning while largely preserving MMLU performance (Sec.6 / Appendix).

In summary, this work transforms SAE from an "interpretation tool" into an "internal OOD thermometer," using the signal for two purposes: improving robustness and fine-tuning efficiency through sample selection, and using jailbreak as OOD for defensive alignment.

**Compliance With Llm Reviewing Policy:**

Affirmed.

**Final Justification:**

After reading the responses, my overall evaluation does not change, and I will keep my current score.

**Key Questions For Authors:**

1. How sensitive is the energy score $F(x)$ to the SAE choice (layer, width, training recipe) and to the thresholding method (z-score vs one-class SVM)? If the signal is stable across reasonable choices, I would rate the method as substantially more sound and reusable.

2. For the LoRA defense, you emphasize jailbreak success dropping (e.g., 46% → 7% on held-out prompts) and note that MMLU changes are tiny, but do not report downstream benign helpfulness, over-refusal, or other behavioral regressions in the main writeup—leaving a key safety tradeoff unquantified. Could you add these key metrics in the context of refusal to respond? (In case your success is largely due to excessive rejection of *Benign Queries*)

**Limitations:**

Because the core OOD setting is largely synthetic, it is unclear how directly the proposed SAE-energy probe transfers to real-world distribution drift. The jailbreak component also raises dual-use concerns: the same pipeline that identifies OOD signatures and “jailbreak-exclusive” concept directions could plausibly be used to build stronger attacks or evade defenses. The paper would benefit from clearer mitigation guidance and from reporting compute cost for SAE extraction and fine-tuning.

**Strengths And Weaknesses:**

**Strengths.** The paper’s core story is coherent: it links distribution shift to representational “off-manifold” signatures measurable via SAE readouts, and uses a deliberately simple perturbation (typos) to make the phenomenon clean and reproducible (Fig.2). Importantly, it does not stop at correlation—it proposes concrete downstream uses: OOD detection thresholds with high overall accuracy at sensible cutoffs (Appendix Table 5), data selection for robust fine-tuning, and a safety-motivated application to jailbreak suppression (Sec.5–6). The attempt to extend beyond text (e.g., CLIP/vision perturbations in the appendix) also helps argue the idea is not purely tokenization-specific.

**Weaknesses.**
(i) The central OOD evidence is dominated by *synthetic surface-form corruption* (typos). I agree that is a good stress test, but it is not representative of many real shifts (domain/style drift, instruction framing, conversational history). The paper would benefit from **at least one “natural” OOD regime** where distribution shift is not merely orthographic noise, and from error-mode analyses linking high energy to specific failure types.

(ii) Robustness fine-tuning relies on an oracle corruption mask. The typo-robustness training masks the *exact* typo positions in the loss so the model “reads typos but does not generate them” . This assumes corruption locations are known during training (often unrealistic), and may materially inflate the apparent gains from energy-based sample selection.

(iii) Practical OOD use appears to require shift-specific tuning/labels.The paper states the “optimal threshold varies” with OOD type and intensity , and selects “optimal threshold (selected on a held-out test set)” when comparing layers . This weakens the implied “drop-in” inference-time diagnostic, since deployment would need calibration for each shift.

---

> ### Author Rebuttal · Authors · 2026-03-31
>
> We thank the reviewer for their thorough reading of our work, and that they found the narrative to be highly coherent and recognize the utility of our method for “defensive alignment” of LLMs. We have addressed the remaining concerns and questions below.
>
> **Weaknesses**
>
> W1: We are in full agreement with the reviewer that surface form corruptions (typos) do not cover the full spectrum of possible OOD inputs. We introduce this toy experiment initially as a highly controllable test bed to isolate the OOD phenomenon as cleanly as possible. In Appendix A.9 we also investigate two additional forms of more natural OOD: the works of Shakespeare and modern poetry (compared with in-distribution TinyStories samples). These would be examples of style drift OOD. Finally, in the main text, we re-frame jailbreaks as an OOD problem, and intervene on the model through SAE-guided fine-tuning to stretch the generalization abilities of Llama 3.1 8B. These would be examples of instruction framing OOD. Finally, in Appendix A.5 we investigate image OOD by swapping image patches. While by no means exhaustive, we believe that these forms of OOD cover a fairly wide spectrum of possible OOD model behaviors.
>
> W2: This is an interesting point. We note that the results are largely the same with/without this typo mask. We chose to employ a typo mask in the final results to make the fine-tuning task more realistic; it would be unnatural to fine-tune an LLM to purposefully produce typos. The typo mask in the loss function ensures that the **activations** of the typo tokens are read by the model, but the next token prediction task does not reward the production of typos. This mask is only applied at the **end** of the token sequence, as generation occurs, and does not apply to **interior** typos that the model has to read. This sort of masking is certainly not necessary or core to our experiments or conclusions; for a different kind of fine-tuning task (such as jailbreaks) there is no need to apply this mask.
>
> W3: In general our method does not rely on shift-specific tuning; rather for the narrow task of OOD **detection** presented in Appendix A.11 we are required to set a threshold for binary classification. See the response to Q1 below for a more thorough explanation of this. In the case of SAE-informed fine-tuning, all that is needed is a straightforward comparison between samples based on how high (OOD) or low (ID) the energy score is for those samples. No need for thresholding.
>
> **Questions**
>
> Q1: The energy score is quite robust to the layer and overall SAE architecture choices. We explicitly evaluate fine-tuning results and energy score distributions at various layers in Appendix A.7. We note that the energy score is only well formed for vanilla SAEs trained with an L1 penalty (i.e. not top-k or batch top-k SAEs). Indeed, this is a deliberate design decision. Still, we notice that the width of the SAE latent space is largely irrelevant; we train SAEs on different random seeds and different latent widths with largely similar results. The pre-trained Llama 3.1 8B SAE also yields consistent results when compared with our toy experiments.
>
> Again, our thresholding methods presented in Appendix A.11 are not strictly necessary to use the energy score, they are only presented for the narrow task of binary OOD detection. As the reviewer correctly notes, OOD is not a binary state, rather it is better described as a continuous spectrum. Therefore, in order to perform binary OOD detection within some dataset, it is necessary to find a detection threshold that is specific to that dataset. Z-scoring or SVM classifiers (or any other single-variable energy score classifier) work equally well, and can be formulated to give identical results. For fine-tuning, we find it preferable to use the energy score as a continuous OOD measurement, and disregard these classifiers.
>
> Q2: We thank the reviewer for this excellent suggestion. We evaluated our SAE-fine-tuned Llama 3.1 8B on a subsample of OR-BENCH, a widely used over-refusal benchmark, and found nearly identical scores: a 3.0% benign refusal rate compared to 3.5% for the base model. Due to computational constraints, we used a string-matching scoring strategy instead of the full OR-BENCH evaluation pipeline. We will add implementation details to the Appendix. We conclude that our targeted fine-tuning hardens jailbreak defenses without compromising model utility.
>
> **Limitations**
>
> We will update the Limitations section to more explicitly mention the wide spectrum of possible OOD inputs that we did not cover. As well, we will provide additional guidance for mitigation of possible attacks using this method. A key piece of this mitigation is robust evaluations and possible red-teaming using our SAE-informed fine-tuning method. We will also more clearly report the compute requirements to pre-train GPT-2 from scratch on TinyStories, train SAEs, and perform inference on the various models.

---

> > ### Author Rebuttal · Reviewer_gcQj · 2026-04-03
> >
> > Overall, the rebuttal addresses several of my concerns, and I am inclined to maintain my original assessment. The main remaining issue is external validity: the core evidence still relies too heavily on synthetic typo-based OOD, and I would most like to see stronger results on more natural, deployment-relevant distribution shifts in future revisions.

---

> > > ### Author Response · Authors · 2026-04-07
> > >
> > > We thank the reviewer for considering our initial rebuttal and their thoughtful additional comments. We do agree with the reviewer that additional results beyond toy OOD settings would serve to strengthen the paper. Therefore, based on this helpful reviewer feedback, we have conducted an additional SAE-guided fine-tuning experiment on an entirely new, naturalistic form of OOD and problem setting.
> > >
> > > For this new experiment, we consider the robustness of Llama 3.1 8B to automatic speech recognition (ASR) transcripts and subsequent question answering based on these transcripts. This is likely a very common instance of distribution shift for LLMs in the wild: extracting key pieces of information from potentially messy transcribed meeting notes or customer service logs. We source these ASR transcripts from Spoken-SQuAD [1], a dataset consisting of ASR-derived context snippets combined with subsequent open-ended questions about these context snippets. These ASR-derived context snippets contain common ASR artifacts such as incorrect words, split words, phonetic misspellings, and grammatical errors. Each sample in Spoken-SQuAD has an analogous sample with clean context in the normal Stanford Question Answering Dataset (SQuAD) [2].
> > >
> > > We consider Spoken-SQuAD samples to be OOD, and normal SQuAD samples to be ID for a vanilla Llama 3.1 8B model. Thus, similar to how we perform SAE-guided fine-tuning for jailbreak resistance in the paper, we fine-tune LoRA adapters to align the SAE concepts of layer 19 between the Spoken-SQuAD and SQuAD datasets. We find that after training on a subset of 5000 examples from the training sets of Spoken-SQuAD/SQuAD, we are able to significantly improve the Q&A abilities of our SAE fine-tuned Llama 3.1 8B model given the slightly garbled ASR-derived contexts from Spoken-SQuAD. We score the generated answers using an exact match (EM) scheme. On 5351 validation set examples for Spoken-SQuAD, we see an exact match accuracy of 58.33% after fine-tuning, compared to only 49.45% on the Spoken-SQuAD validation set with the base LLM. Even with these OOD context sections, the SAE-guided fine-tuning strategy approaches the exact match accuracy of 59.97% for the base LLM on the in-distribution SQuAD validation set. In fact, using only the SAE features as supervisory signals, the SAE fine-tuned LLM demonstrates better performance on the normal SQuAD eval set than the base LLM, with an exact match accuracy of 67.88%. This shows that our SAE-guided fine-tuning approach appears to improve extractive Q&A across context distributions. We will add these new results along with more experimental details to the paper.
> > >
> > > **Spoken-SQuAD/SQuAD validation set accuracy**
> > > |  | Q&A Accuracy (EM, %) |
> > > |---|---|
> > > | Base Llama 3.1 8B (Spoken-SQuAD val.) | 49.45 |
> > > | SAE fine-tuned Llama 3.1 8B (Spoken-SQuAD val.) | 58.33 |
> > > | Base Llama 3.1 8B (SQuAD val.) | 59.97 |
> > > | SAE fine-tuned Llama 3.1 8B (SQuAD val.) | 67.88 |
> > >
> > > We thank the reviewer for suggesting these additional analyses. We are confident that this new, highly deployment-relevant use-case provides an excellent showcase for the implications of the conceptual and practical frameworks introduced in the paper.
> > >
> > > We now present OOD cases ranging from single-character typos (in the toy setting with character-level tokenization), to style shifts between TinyStories, the works of Shakespeare, and modern poetry, to jailbreaks, and now to ASR transcripts with extractive Q&A.
> > >
> > > [1] Li et al., 2018. Spoken SQuAD: A Study of Mitigating the Impact of Speech Recognition Errors on Listening Comprehension
> > >
> > > [2] Rajpurkar et al., 2016. SQuAD: 100,000+ Questions for Machine Comprehension of Text

---

### Official Review · Reviewer_yKLG · 2026-03-13

**Soundness:** 3
**Presentation:** 4
**Significance:** 3
**Originality:** 4
**Overall Recommendation:** 5
**Confidence:** 4

**Summary:**

The authors are trying to use sparse autoencoders to understand what transformers do on OOD data. In order to understand OOD data, we need to understand what's representable by the model at the neuronal level. Given the linear representation hypothesis, we can study the accumulation of feature data through a transformer architecture via the residual stream activation which is approx $x = \sum_i \alpha_i f_i$ where $f_i$ are learned features and $\alpha_i$ are the corresponding learned feature coefficients. Goal is to understand the representational capacity of the model by understanding the features $f_i$ and their coefficients $\alpha_i$. A good way to do this is to model the residual stream with a sparse autoencoder. Given the sparse autoencoder, the authors construct an energy score which is error for SAE reconstruction plus penalty for number of features used in the reconstruction. Typos are an example of out-of-distribution data, so authors train a toy model on TinyStories, then test for OOD-ness with injected typos. They also study the features activated by jailbreak prompts and are able to finetune models with an awareness of the feature manifold to be more robust to these jailbreaks.

**Compliance With Llm Reviewing Policy:**

Affirmed.

**Final Justification:**

As noted in my rebuttal, no update to my score needed.

**Key Questions For Authors:**

- Why choose that exact form for the energy score?
- How well does the model do on other modalities, like out of distribution images? How about on adversarially attacked images, like Goodfellow's "panda to gibbon"? https://arxiv.org/pdf/1412.6572
- Were you able to calculate the residual stream for the frontier models?

**Limitations:**

Yes

**Strengths And Weaknesses:**

Strengths:
- Very well contextualized in its literature, with lots of different threads coming together to motivate this paper and what authors are seeking to do
- Jailbreak work is very impressive, and seems like a novel advance in robust training of transformer architectures. It gives much insight into what's happening in the jailbreaking phenomenon

Weaknesses:
- I'm not sure if one-character typos are significant enough OOD to believe that this will generalize to even more OOD settings --- OOD-ness is a spectrum!
- Demonstrating in Figure 2 that typos reduce LLM performance feels repetitive of Gan et al (2024)
- More discussion of the chosen energy score, which seems to be a combination of autoencoder error and an information-theoretic penalty, which would have a deep literature. In particular, this paper could be truly great with a discussion of the asymptotic properties of this energy score
- Anomaly detection with autoencoders is well studied, and the authors could have referenced some of this literature. There are also pitfalls to the use of autoencoders for anomaly detection: see results in https://arxiv.org/pdf/2501.13864
- Figure 1b caption refers to a shaded region, but there is no shaded region in figure 1b

---

> ### Author Rebuttal · Authors · 2026-03-31
>
> We thank the reviewer for their positive appraisal of our work, and for noting the novelty of the approach and its practical uses. We have addressed the reviewer comments and key questions below.
>
> **Weaknesses**
>
> W1: We fully agree with the reviewer that OOD-ness is a (potentially complex) spectrum. To partially account for this, we provide a gradation of typo frequency from 0% of words with typos to 100% of words containing a single-character typo. Still, we understand that this is just a single manifestation of OOD. Therefore, in Appendix A.9, we investigate OOD in the form of writing style, with Shakespeare and modern poetry as two natural variants compared with an LLM trained purely on the TinyStories corpus. Further, we frame successful jailbreaks as an OOD phenomenon and mitigate their effectiveness with SAE-targeted fine-tuning. While this is by no means an exhaustive survey, we investigate OOD from simple (single-character typos) to complex (sophisticated jailbreaks) scenarios.
>
> W2: We agree that our post-typo benchmark performance degradation results are not by themselves novel. Our aim was to pair these results (Figure 2a) with the activation of additional concepts (Figure 2b) from OOD inputs, linking the model’s internal state with observed deficits under OOD. We acknowledge that the current framing may oversell the novelty of the performance results in isolation. We have accordingly revised the Introduction and Abstract to de-emphasize these results alone and have added citations to Gan et al. and other prior work where appropriate.
>
> W3: This is an excellent suggestion. We do have an information theoretic motivation for constructing this energy score; we will explain this in more detail in the Methods section. Our motivation for introducing this energy score metric was to combine two complementary notions of “informativeness” derived from the SAE.
>
> The first aspect corresponds to the number of semantic concepts (the SAE L0) identified in the residual stream activations of a transformer model layer. These concepts are linearly disentangled “directions” in SAE space that describe the sample. From an information-theoretic perspective, this “concept count” is similar to the minimum description length. If a sample is out of the ordinary (i.e., induces an OOD internal processing stream), the SAE likely requires more (and rarer) concepts to “explain” this atypical representation.
>
> The second aspect corresponds to the reconstruction error between the SAE reconstruction and the true residual stream representation. Even with extra concepts for OOD samples, if the sample lies too far from the LLM representational manifold, the SAE as a linear approximator cannot sufficiently reconstruct the true activations from the sparse concepts alone.
>
> W4: We acknowledge the reviewer’s point about existing work on anomaly detection with autoencoders. There are certainly some similarities between this research area and our work, particularly if OOD data is viewed as “anomalous”. A key point of differentiation is our use of overcomplete *sparse* autoencoders instead of bottleneck autoencoders. While bottleneck AEs rely on compressed latent spaces and reconstruction error, our approach uses an overcomplete basis to decompose dense activations into monosemantic units, enabling more targeted interventions and avoiding unreliable ID/OOD separation. We instead use extraneous concept activation as the primary OOD signal. We will include a clearer contrast with anomaly detection AEs in the Related Work section.
>
> W5: We thank the reviewer for pointing this out. There is a very thin shaded region in Figure 1b indicating 1 standard deviation of excess SAE feature activations across 50 random typo configurations, but it is almost imperceptible due to the stability across configurations. We will update the figure with a more visible representation of this variance.
>
> **Questions**
>
> Q1: See our response to W3 on the information theoretic motivations for this form of the energy score. We note that there may be other, perhaps more performant, OOD detection metrics that could be derived from SAEs. We explore the current form of the energy score due to its clear motivation and ease of computation.
>
> Q2: This is an excellent question. In Appendix A.5, we explore OOD characterization for ViTs, and find that additional concepts are activated to an even greater extent for OOD images (constructed by patch-swapping portions of the image) compared to text. We do not look at explicitly adversarial images, but we are confident that such images would show an SAE-detectable signature of a spike in spurious concepts “confusing” the model to an adversarial classification. This is a very interesting direction for future work.
>
> Q3: Without access to the activations or residual streams of frontier models, we unfortunately can not report interpretability results for them. The largest model we study mechanistically is a pre-trained Llama 3.1 8B.

---

> > ### Author Rebuttal · Reviewer_yKLG · 2026-04-02
> >
> > My concerns have been resolved. I really like this paper and look forward to any revisions that discuss the energy score further. Otherwise my score is good to remain where it is as a 5.

---

> > > ### Author Response · Authors · 2026-04-07
> > >
> > > We thank the reviewer for their positive feedback and for their support of our work. We are glad to hear that our rebuttal addressed their concerns. In the final version of the manuscript, we will expand the discussion of the energy score to incorporate the points raised during the review process.

---

### Official Review · Reviewer_NS9E · 2026-03-13

**Soundness:** 2
**Presentation:** 3
**Significance:** 3
**Originality:** 3
**Overall Recommendation:** 4
**Confidence:** 3

**Summary:**

The authors use SAEs to understand how OOD data is processed by the model. They do this on a toy model trained on TinyStories with typos as OOD input, as well as study how fine-tuning models based on data scored by an SAE-based "energy score" helps the model better generalize to OOD typos.

**Compliance With Llm Reviewing Policy:**

Affirmed.

**Final Justification:**

The paper is quite sound in the toy setting but it would be good to have more work outside of the toy model setting.

**Key Questions For Authors:**

1. Section 4.1: L256 — Is this monotonically/linear growth true for all layers, other than just layer 6?
2. Section 4.1: Also, you mention that other than number of features activating, the second aspect is the reconstruction error increasing — is this true for your toy model too?
3. Section 4.2: Is your MMLU over all examples?
4. Section 4.2: If so, can you make sure that the MMLU degradation is not due to causally important degradation in the question? e.g. If you have a math question "Find the difference of 142.76 – 16.5." and you perturb any of the character in the number words, this would meaningfully change the question.
5. Section 4.2: Also, some MMLU questions are quite short (e.g. less than 10 words), how would the 5% work here? Is it a 5% probability for each word to be perturbed, or for each sample you designate at least 5% perturbed but with a random choice of which 5%? Do you round up to at least one word?
6. Section 6: I note that the jailbreak-finetuning works in practice, but I'm not sure what kinds of conclusions to draw from it. For example, how does this compare to normal safety finetuning? I would be hesitant to perform the latent finetuning as proposed as I am not sure about the side-effects as compared to more traditional methods.
7. L410: You observe higher raw L0 but what about reconstruction error?

Suggestion: Expand this study to better understand model-OOD vs SAE OOD: Include situations where (1) model is trained with typos and SAE is not, and (2) model is trained without typos and SAE is.

**Limitations:**

yes

**Strengths And Weaknesses:**

**Strengths**
* (Soundness) By first training a toy model on TinyStories, it is a useful way to get ground truth on OOD-ness.
* (Significance, Originality) I think it is an interesting idea to identify OOD shifts by SAEs. I find the energy score defined to be potentially quite useful (e.g. for their fine-tuning experiments).

**Weaknesses**
* (Soundness) It is not obvious whether the findings are due to model-OOD, SAE-OOD, or both. This could be established with further work done in the toy-model domain. As such, I am unsure about the conclusions that could be made in the jailbreaks experiments (whether these are because jailbreaks are OOD for model, SAE, or both)
* (Soundness) I am also not sure the extent how arbitrary the energy score is. Are both true? (e.g. see my second question on whether the recon error is true for the toy model and my last question on whether the recon error is true for the jailbreaks). I note that it somewhat works in practice but more work should be done on making this energy score more principled and stress-tested.
* (Originality) Contribution number 2 is not novel, but this did not significantly affect my score. Input perturbation is just generally quite well studied in the NLP/LLM domain. (e.g. see https://www.nature.com/articles/s41598-025-29770-0)
* (Presentation) Minor, but sections 4 onwards could be better structured with some subsections.

---

> ### Author Rebuttal · Authors · 2026-03-31
>
> We thank the reviewer for their thorough feedback. We have addressed their outstanding concerns below.
>
> **Weaknesses**
>
> W1: This is a very subtle and interesting point. By design, the SAE is trained to reconstruct the activations of the LLM, and the SAE is fit over the LLM activation space. We treat the SAE as a sparse linear approximator of the underlying LLM representational manifold, and therefore use it as a proxy to probe OOD behavior in the LLM. In practice, SAEs are trained on a subset of an LLM's pretraining data. We used a controlled toy model trained on the exact same dataset as the SAE to ensure their data distributions were equivalent; thus, any data OOD for the LLM was also OOD for the SAE. We extend these findings to larger models, suggesting that jailbreaks are OOD for both the LLM and the SAE. We appreciate the reviewer’s suggestion, but since an SAE is definitionally a map of the LLM representation space, training it on a distribution the LLM has never encountered yields trivial reconstruction errors that reflect a fundamental lack of feature alignment between the SAE and the LLM, which we indeed observe.
>
> W2: We introduced this energy score metric to combine two complementary notions of “informativeness” derived from the SAE. The first aspect is the number of semantic concepts (the SAE L0) identified in a transformer layer's residual stream activations. These concepts are linearly disentangled “directions” in SAE space that describe the sample, conceptually similar to the minimum description length. If a sample is out of the ordinary (inducing an OOD internal processing stream), the SAE likely requires more, and rarer, concepts to “explain” this atypical representation. The second aspect is the reconstruction error between the SAE reconstruction and the true residual stream representation. Even with extra concepts for OOD samples, if the sample lies too far from the LLM representational manifold, the linear SAE cannot sufficiently reconstruct the true activations from sparse concepts alone.
>
> It is likely possible to construct a superior SAE-derived OOD metric. We view our main contribution as demonstrating the overall utility of SAEs for mapping the generalization abilities of LLMs, and our energy score is an example of a practical metric that is motivated by these findings.
>
> W3: We fully agree that these typo OOD results are not novel by themselves. In the paper, our aim was to pair these performance results (Figure 2a) directly with the corresponding activation of additional concepts (Figure 2b) from these OOD inputs, thereby linking the internal state of the model with tangible model deficits in the presence of OOD. As such, we have amended our statement of core contributions in the Introduction and Abstract to de-emphasize the benchmark performance results on their own. We have also cited the relevant works from on this topic where appropriate.
>
> W4: We appreciate the reviewer's feedback. We have added subsections to the results sections for improved readability.
>
> **Questions**
>
> Q1. We do not observe explicitly linear growth at all layers. Early layers operate in a syntactic regime where inputs remain ID, resulting in stable concept activation regardless of typo levels. Activation of spurious concepts is most pronounced in middle-late layers during semantic inference. We will add new plots to the Appendix to illustrate this trend for all layers.
>
> Q2. Yes, increased SAE reconstruction error increase with increased OOD is observed in the toy experiments as well, although the effect is less pronounced than with the L0.
>
> Q3. All instances of the MMLU evaluation in this study use the entire test set. We will make this more clear where MMLU results are reported.
>
> Q4. We appreciate this observation. To ensure that we don’t perturb the critical parts of the question, such as numerals, we avoid inducing typos in numerals specifically. We will add this detail to our “typo recipe” in Appendix A.6.
>
> Q5. This is a useful clarification. We corrupt a fixed percentage of words per sample, causing extremely short questions to "round up" to at least one typo. These cases represent only ~5% of the MMLU dataset, likely having minimal impact on the results.
>
> Q6. To show that SAE-guided fine-tuning enables targeted and significant robustness improvements, we note that MMLU performance remains virtually unchanged. In response to reviewer gcQj, we find that the model is unaffected on benign refusal benchmarks (OR-BENCH) and in response to reviewer gmns, we find natural language jailbreak detectors only identify 48.3% of harmful jailbreaks.
>
> Q7. We also observe slightly higher SAE reconstruction error for successful jailbreaks, but it is much less separable compared to SAE L0. This is likely because jailbreak OOD for pre-trained Llama 3.1 8B is a less extreme form of OOD than the typo OOD for the toy setting. We will include a histogram of SAE reconstruction error for the jailbreak dataset in Appendix A.8.

---

> > ### Author Rebuttal · Reviewer_NS9E · 2026-04-04
> >
> > I thank the authors for pointing out that since the SAEs were trained on the exact same dataset in the toy-model setting, this would point to model OOD-ness. I think generally this makes the toy model setting quite a good, strong contribution. However I agree with reviewer gcQJ that there is still a gap with regards to external validity. In light of this I have increased my score to a 4.

---

> > > ### Author Response · Authors · 2026-04-07
> > >
> > > We thank the reviewer for considering our initial rebuttal and for their helpful additional comments. We do agree with the reviewer that additional results on more naturalistic and deployment-critical OOD settings would serve to strengthen the paper. Therefore, based on this helpful reviewer feedback, we have conducted an additional SAE-guided fine-tuning experiment on an entirely new, naturalistic form of OOD and problem setting.
> > >
> > > For this new experiment, we consider the robustness of Llama 3.1 8B to automatic speech recognition (ASR) transcripts and subsequent question answering based on these transcripts. This is likely a very common instance of distribution shift for LLMs in the wild: extracting key pieces of information from potentially messy transcribed meeting notes or customer service logs. We source these ASR transcripts from Spoken-SQuAD [1], a dataset consisting of ASR-derived context snippets combined with subsequent open-ended questions about these context snippets. These ASR-derived context snippets contain common ASR artifacts such as incorrect words, split words, phonetic misspellings, and grammatical errors. Each sample in Spoken-SQuAD has an analogous sample with clean context in the normal Stanford Question Answering Dataset (SQuAD) [2].
> > >
> > > We consider Spoken-SQuAD samples to be OOD, and normal SQuAD samples to be ID for a vanilla Llama 3.1 8B model. Thus, similar to how we perform SAE-guided fine-tuning for jailbreak resistance in the paper, we fine-tune LoRA adapters to align the SAE concepts of layer 19 between the Spoken-SQuAD and SQuAD datasets. We find that after training on a subset of 5000 examples from the training sets of Spoken-SQuAD/SQuAD, we are able to significantly improve the Q&A abilities of our SAE fine-tuned Llama 3.1 8B model given the slightly garbled ASR-derived contexts from Spoken-SQuAD. We score the generated answers using an exact match (EM) scheme. On 5351 validation set examples for Spoken-SQuAD, we see an exact match accuracy of 58.33% after fine-tuning, compared to only 49.45% on the Spoken-SQuAD validation set with the base LLM. Even with these OOD context sections, the SAE-guided fine-tuning strategy approaches the exact match accuracy of 59.97% for the base LLM on the in-distribution SQuAD validation set. In fact, using only the SAE features as supervisory signals, the SAE fine-tuned LLM demonstrates better performance on the normal SQuAD eval set than the base LLM, with an exact match accuracy of 67.88%. This shows that our SAE-guided fine-tuning approach appears to improve extractive Q&A across context distributions. We will add these new results along with more experimental details to the paper.
> > >
> > > **Spoken-SQuAD/SQuAD validation set accuracy**
> > > |  | Q&A Accuracy (EM, %) |
> > > |---|---|
> > > | Base Llama 3.1 8B (Spoken-SQuAD val.) | 49.45 |
> > > | SAE fine-tuned Llama 3.1 8B (Spoken-SQuAD val.) | 58.33 |
> > > | Base Llama 3.1 8B (SQuAD val.) | 59.97 |
> > > | SAE fine-tuned Llama 3.1 8B (SQuAD val.) | 67.88 |
> > >
> > > We thank the reviewer for suggesting these additional analyses. We are confident that this new, highly deployment-relevant use-case provides an excellent showcase for the implications of the conceptual and practical frameworks introduced in the paper.
> > >
> > > We now present OOD cases ranging from single-character typos (in the toy setting with character-level tokenization), to style shifts between TinyStories, the works of Shakespeare, and modern poetry, to jailbreaks, and now to ASR transcripts with extractive Q&A.
> > >
> > > [1] Li et al., 2018. Spoken SQuAD: A Study of Mitigating the Impact of Speech Recognition Errors on Listening Comprehension
> > >
> > > [2] Rajpurkar et al., 2016. SQuAD: 100,000+ Questions for Machine Comprehension of Text

---

### Official Review · Reviewer_gmns · 2026-03-13

**Soundness:** 3
**Presentation:** 3
**Significance:** 2
**Originality:** 2
**Overall Recommendation:** 4
**Confidence:** 3

**Summary:**

The authors apply SAEs trained on residual stream activations as an OOD input classifier. The authors show that OOD inputs (typos, style shifts, jailbreaks) cause the SAE to activate more latent features and incur higher reconstruction error. They unify these findings into a single "energy score." They demonstrate that even minor typos degrade benchmark performance across models from GPT-2 to GPT-5-thinking-nano, and that this degradation tracks the expansion in activated SAE features. They then show that the energy score can prioritize high-value OOD samples for more efficient fine-tuning and apply the framework to jailbreak defense, where LoRA-based alignment of SAE feature activations reduces jailbreak success from Llama. This improvement in ASR does not incur a steer safety tax.

**Compliance With Llm Reviewing Policy:**

Affirmed.

**Final Justification:**

Thank you to the authors for this response. I feel I have a clearer sense of the motivation for this work, and am glad to see the authors more clearly note its limitations. I have raised my score.

**Key Questions For Authors:**

1. Do the authors consider this work to contribute more to the basic science of SAEs or as a practical intervention? If the latter, I'd recommend that the authors broaden their experimental scope. This includes a more diverse suite of models across parameter scales, other SAe architectures, and more rigorous multi-turn jailbreak attacks.

2. The paper repeatedly describes extra SAE features activated on OOD inputs as "spurious" and "distracting," but never formally defines or validates this characterization. Do the authors consider classifying these features as such a load-bearing part of this paper's conceptual findings? If so, I'd be excited to see a clearer definition/example.

**Limitations:**

This paper would benefit from more directly engaging with its limitations. The most direct engagement is limited to a few sentences at the end, lacking nuance and detail. I'd be excited to see empirical validation that this intervention holds up across a diverse set of SAEs rather than ***"Our approach assumes a sufficiently well-trained SAE (high explained variance and appropriate sparsity)"***.

**Strengths And Weaknesses:**

**Soundness**: The experimental setup seems broadly reasonable. However, many of the core conceptual claims rely on many of the features (aka SAE latents) being spurious. I was unable to glean a clear definition of which features the authors consider spurious. This ambiguity is not improved by the confident claims in the abstract about the extra features being spurious, followed by hedging language (***"potentially spurious"***) in the paper's body (e.g., line 81). I'm also unsure why the authors opted to leverage closed-weight LLMs offered by a private company instead of open-source alternatives that they could directly apply their SAE methodology to.

**Presentation**: The paper is generally well-written, with a clear narrative arc from toy experiments to real-world applications. The figures are informative and well-constructed, particularly Figures 1 and 4, which effectively communicate the core findings. The progression from controlled typo experiments through benchmark evaluation to jailbreak defence provides a logical structure.

**Significance**: This work contributes to the growing literature on defences against jailbreak attacks and on LLM reliability overall. These are crucial problems. Guarding against misuse and ensuring reliability is essential if society is to realize the benefit of transformative AI while mitigating its risks. This work focuses on the more abstract OOD problem, which can encompass reliability and misuse/jailbreaks. However, I'd be excited for the authors to make a case that SAEs outperform existing jailbreak defences, including auxiliary natural language input classifiers. While not necessary if the authors are satisfied with contributing to the basic science of SAEs, such a case would make the practical counterfactual impact of this intervention clearer.

**Originality**: The finding that minor changes to evaluation questions can lead to surprising degradations in LLM performance is unsurprising [1, 2]. The current paper's narrative gives the impression that this is a novel insight. I suggest that the authors moderate their claims and instead focus on the SAE OOD detection.

[1] - Gan, E., Zhao, Y., Cheng, L., Yancan, M., Goyal, A., Kawaguchi, K., ... & Shieh, M. (2024, November). Reasoning robustness of llms to adversarial typographical errors. In Proceedings of the 2024 Conference on Empirical Methods in Natural Language Processing (pp. 10449-10459).

[2] - Nishant Balepur, Rachel Rudinger, and Jordan L. Boyd-Graber. Which of these best describes multiple choice evaluation with llms? a) forced b) flawed c) fixable d) all of the above. ArXiv, abs/2502.14127, 2025.

---

> ### Author Rebuttal · Authors · 2026-03-31
>
> We thank the reviewer for their constructive comments, and appreciate that they found our paper to be well-written and the experiments to be reasonable. We have addressed each of their concerns below.
>
> **Soundness**
>
> We acknowledge that our use of "spurious" may be under-motivated. We find OOD inputs consistently activate more latent concepts than semantically equivalent in-distribution inputs (e.g., following induced typos). These additional concepts are not semantically relevant but are necessary to represent off-manifold OOD inputs. This is more evident in the image domain, where ViTs activate extraneous concepts not present in the image (e.g., a "sailboat" concept for a cat). We perform OOD vision experiments in Appendix A.5 and include examples there to clarify why these concepts are likely "spurious." Indeed, recent literature suggests these concepts can act as distractors leading to hallucinations [1]. Our work does not hinge on these features being "spurious"; while some may have semantic relevance, they appear largely irrelevant and induced primarily by the OOD nature of the inputs.
>
> Figure 2b evaluates surface-form perturbations (typos) across models ranging from Llama 3.1 8B (open) to GPT-5-nano (closed), demonstrating that these OOD effects persist even in powerful reasoning models. We then applied our SAE methodology to Llama 3.1 8B and GPT-2. This analysis shows that such behaviors are not mere capability gaps that evaporate with scale, but are intrinsic to LLM architectures and training regimes. Having established these stakes, we focus exclusively on open-weight models for the remainder of the paper.
>
> **Significance**
>
> As noted by the reviewer, our focus was LLM behavior under OOD, using jailbreaking as a consequential proof point. While our main contribution centers on the basic science of LLMs and using SAEs to characterize internal manifolds, we agree that comparing our method with external classifiers is compelling. We therefore benchmarked our SAE-based method against Llama Prompt Guard 2 on the WildJailbreak evaluation set. Llama Prompt Guard 2 identifies only 48.3% of adversarial jailbreaks as harmful, indicating that auxiliary natural language classifiers are insufficient and underscoring the need for targeted hardening of base LLMs. More comprehensive results will be added to the Appendix.
>
> **Originality**
>
> We agree that small typographic perturbations degrading benchmark performance in frontier LLMs is not novel by itself. Our contribution instead frames this effect within OOD behavior: such perturbations both reduce benchmark performance (Figure 2a) and trigger additional concept activation in the model’s representation space (Figure 2b). We aim to link these as joint OOD phenomena: unexpected behavior alongside elevated concept activation relative to the baseline inputs. We acknowledge that the current framing may overstate the novelty of the performance results alone. In response to the reviewer feedback, we revised the Introduction and Abstract to de-emphasize this aspect, moderated claims in Section 4.2, and added citations to better contextualize our findings.
>
> **Limitations**
>
> We acknowledge that our discussion of the limitations of our work could be more nuanced. Empirical validation across diverse SAEs requires high explained variance and optimal sparsity, as evaluated in the literature. Improperly trained SAEs with poor manifold coverage if the LLM representation space may incorrectly flag ID inputs as OOD, while insufficient sparsity risks the model simply learning the identity function. As noted in our response to Reviewer NS9E, we ensured that our SAEs cover the entire manifold of the toy model to prevent misleading signals. Our investigation spans a diverse array of SAEs, including various random seeds and latent widths, all yielding consistent results. These findings are further supported by our results using the pre-trained Llama 3.1 8B SAE on the jailbreaking task. Because our framework relies on unbounded concept activation counts (L0), we intentionally utilize vanilla L1 SAEs rather than top-K architectures. We will elaborate on these technical constraints in the revised manuscript.
>
> **Questions**
>
> Q1: We intend this paper to focus more on the basic science of SAEs as tools for understanding LLM behavior and failure modes. While our framework has practical implications for sample-efficient fine-tuning and jailbreak prevention, we do not exhaustively catalogue all applications, and leave further exploration to future work.
>
> Q2: Classifying these features as "spurious" is not core to our theoretical framework, but rather a labeling choice. While we are open to more appropriate labels for these "extra" features, our work in the image domain suggests many are semantically irrelevant to the base inputs and could, therefore, be considered "spurious."
>
> [1] Suresh et al., NeurIPS 2025. From Noise to Narrative: Tracing the Origins of Hallucinations in Transformers.

---

> > ### Author Rebuttal · Reviewer_gmns · 2026-04-04
> >
> > Thank you to the authors for this response. I feel I have a clearer sense of the motivation for this work, and am glad to see the authors more clearly note its limitations. I have raised my score.

---

> > > ### Author Response · Authors · 2026-04-07
> > >
> > > We thank the reviewer for their careful consideration of our rebuttal, and we are encouraged that our responses have adequately addressed their concerns.

---

### Decision · Program_Chairs · 2026-04-30

**Decision:**

Accept (regular)

**Comment:**

Thanks for your submission to ICML 2026. This submission proposes an interesting finding: a metric "energy score" can be derived from sparse autoencoders to trace the out-of-distribution degree, verified on synthetized typo data and natural jailbreak data. Moreoever, the SAE discovered concept can be intentionally aligned to enhance LLM alignment against jailbreak attacks and LLM OOD robustness.

All reviewers and the author were actively involved during the discussion phase. After the discussion, all reviewers appreciate the submission due to the coherent study between SAE alignment and OOD and its strong practical implications. The energy score may of significant potential and demonstrates how SAE improves model robustness/alignment in practice. All reviewers reach edthe consensus of acceptance. Please make sure to incorporate reviewers comments, especially the discussion on the extensibility to settings beyond the typo task, the reliance on well-trained OOD and potential confounding impact from the SAE OOD, along with the design choices of the energy score and its connection with information theory.